# Validation of the Group Environment Questionnaire (GEQ) in a Simulated Learning Environment

**DOI:** 10.3390/nursrep15050154

**Published:** 2025-04-30

**Authors:** José Manuel García-Álvarez, Alfonso García-Sánchez, Alonso Molina-Rodríguez, María Suárez-Cortés, José Luis Díaz-Agea

**Affiliations:** 1Health Sciences PhD Program, Catholic University of Murcia (UCAM), Campus de los Jerónimos nº 135, Guadalupe, 30107 Murcia, Spain; 2Faculty of Nursing, Catholic University of Murcia (UCAM), Campus de los Jerónimos nº 135, Guadalupe, 30107 Murcia, Spain; agarcia@ucam.edu; 3Faculty of Nursing, University of Murcia, Campus de Ciencias de la Salud, Edificio LAIB/DEPARTAMENTAL, El Palmar-Murcia, 30120 Murcia, Spain; alonso.molina@um.es (A.M.-R.); maria.suarez@um.es (M.S.-C.); agea@um.es (J.L.D.-A.)

**Keywords:** group cohesion, psychometric properties, questionnaire, nursing, simulated learning environment

## Abstract

**Background/Objectives**: The complexity of modern healthcare requires teamwork. Healthcare teams must be cohesive to ensure efficient and quality care. The objective of this study was to validate the Spanish short version of the Group Environment Questionnaire (GEQ) in undergraduate nursing students undergoing clinical simulation training. **Methods**: The psychometric properties of the questionnaire were analyzed. We began with a statistical analysis of the items. Internal consistency was assessed using McDonald’s omega coefficient. Test-retest reliability was determined using Spearman’s correlation coefficient. An exploratory factor analysis was performed using the principal components analysis method with varimax rotation. Finally, a confirmatory factor analysis of the proposed theoretical models was performed to select the most appropriate one using the weighted least squares method adjusted for mean and variance (WLSMV) and goodness-of-fit indices. **Results**: The questionnaire items showed a standard deviation higher than 1 and a negative skewness lower than 0.5. Internal consistency and test-retest reliability were higher than 0.8. The item-total correlation coefficient values were above 0.44. The exploratory factor analysis confirmed the presence of four factors, each with three items. Confirmatory factor analysis determined that the four-factor cohesion model was the one that obtained the best fit. **Conclusions**: The Spanish short version of the Group Environment Questionnaire (GEQ) is a valid and reliable instrument for analyzing group cohesion in teams of nursing students undergoing clinical simulation training.

## 1. Introduction

Teamwork among healthcare professionals improves patient safety, reduces stress, increases healthcare staff satisfaction, optimizes resources, improves planning, and reduces costs, increasing the efficiency and performance of healthcare. Wolking as a team allows each member’s individual strengths to be amplified, making it easier to achieve established goals. It is essential to learn how to work in a team by practicing effective communication, developing empathy, respecting others’ ideas, sharing responsibilities, and valuing each member’s contributions [1,2,3,4].

In addition to scientific and technical skills, the healthcare team must possess non-technical skills including communication, coordination, complementarity, trust, and commitment. These non-technical skills are essential for achieving team cohesion and ensuring success in a stressful work environment characterized by changes in technology, care complexity, excessive workload, exposure to patient suffering, and the constant fear of making mistakes [2,5,6].

Group cohesion can be defined as a dynamic process that keeps the team united in the achievement of common objectives and in the satisfaction of the affective needs of its members. Among the different conceptual models developed for the analysis of group cohesion, those that differentiate between task cohesion and social cohesion stand out. Task cohesion refers to the level of commitment and collaboration of team members to achieve the established objectives [7,8,9]. Social cohesion refers to the degree of interaction and camaraderie among team members, fostering personal relationships and satisfaction with the team’s work [7,8]. Both aspects can be approached from a group and individual perspective, giving rise to the four dimensions that make up group cohesion: the Group Integration-Task (GI-T) or degree of group union to achieve common objectives, Group Integration-Social (GI-S) or degree of group union to develop social relations within the group, Individual Attractions to Group-Task (ATG-T) or individual motivations towards common objectives, and Individual Attractions to Group-Social (ATG-S) or individual motivations towards social relations within the group [7,8,9,10,11].

The Group Environment Questionnaire (GEQ) allows the assessment of these four dimensions of group cohesion. The original questionnaire is composed of 18 items with a nine-choice Likert-type response [12,13]. Different adaptations and validations of the GEQ have been carried out for the Spanish language and context. Among them, the short versions consisting of 12 items that have been adapted and validated for professional athletes [14] and for working groups of Spanish university students of Sports Science and Pedagogy [15] stand out. Both versions presented acceptable values of internal consistency both globally and in their different dimensions and adequate values of the adjustment indexes to the four-dimensional model of group cohesion originally proposed [14,15].

Due to the importance of group cohesion in the quality of healthcare based on teamwork and the lack of validated instruments to analyze group cohesion in the Spanish healthcare field, it is necessary to validate the reduced Spanish version of the GEQ in teams of Spanish students and healthcare professionals. For the validation of this questionnaire, nursing students undergoing clinical simulation training were chosen because of the capacity of this learning tool to develop the nontechnical skills needed to improve group cohesion in teams of students and healthcare professionals [16,17].

The objective of this study was to validate the Spanish short version of the GEQ in undergraduate nursing students undergoing clinical simulation training.

## 2. Materials and Methods

### 2.1. Participants

A descriptive and cross-sectional study was carried out to validate the Spanish adaptation of the short version of the GEQ in students of the Nursing Degree of the Catholic University of Murcia (UCAM) and the University of Murcia (UMU), located in the region of Murcia, Spain. The inclusion criteria were to be a fourth-year student of the Nursing Degree who had completed all the clinical simulation sessions and who wished to participate voluntarily in the study. The sample was selected using non-probabilistic convenience sampling based on the groups assigned by the Nursing Practice Unit of each of the participating universities.

### 2.2. Procedure

To collect the information, a questionnaire including sociodemographic data and the short Spanish version of the GEQ was completed in the clinical simulation classrooms of the participating universities between October 2023 and July 2024.

### 2.3. Ethics Statement

The study was conducted taking into account the principles established by the Declaration of Helsinki and with the approval of the Ethics Committee of UCAM and the authorization of the UMU.

Students were informed in detail about the procedures and the nature of the study. It was indicated that participation was voluntary and that they could withdraw from the study at any time. No incentives or rewards were offered for participation and there were no sanctions for non-participation. Written informed consent was obtained from all participants. Confidentiality and anonymity were ensured by not recording any identifying information about the participating students.

### 2.4. Measuring Instrument

The instrument used to measure group cohesion was the Spanish short version of the GEQ validated for groups of Spanish university students of Sports Science and Pedagogy. This questionnaire was composed of 12 items with a five-choice Likert-type response format ranging from totally disagree (1) to totally agree (5). Items 1, 3, and 5 assessed the dimension ATG-S, items 2, 4, and 6 analyzed the dimension ATG-T, items 7, 9, and 11 assessed the dimension GI-S, and items 8, 10, and 12 analyzed the dimension GI-T [15].

The questionnaire items used in this study were rigorously adapted by the researchers to the academic context of Spanish nursing students to preserve content validity in the new context. Specifically, the following terms were substituted: team for group, work for preparation of the simulation sessions, work sessions for conduct of the simulation sessions, and project for clinical simulation (Table 1).

### 2.5. Variables

The sociodemographic variables used in this study were university, age, gender, region of origin, previous academic degree, and work activity. The main variables of this research were the items and dimensions of the questionnaire to assess group cohesion used in this research.

### 2.6. Data Analysis

The information obtained was analyzed using SPSSv26^®^ and AMOS v25^®^ statistical software for Windows (Armonk, NY, USA: IBM Corp.).

A descriptive statistical analysis of the sociodemographic variables was carried out by calculating the frequencies and percentages of the qualitative variables, as well as the mean, standard deviation, skewness, and kurtosis of the quantitative variables. An inferential statistical analysis of the sociodemographic variables was performed by applying different statistical tests according to the characteristics of the variables analyzed [18].

In order to evaluate the psychometric properties of the Spanish short version of the GEQ in undergraduate nursing students undergoing clinical simulation training, a statistical analysis of the items, reliability assessment, and evaluation of construct validity was carried out [19]. For the statistical analysis of the questionnaire items, the following statistical measures were calculated: frequency and percentage, mean, median, mode, standard deviation, skewness, kurtosis, and type of distribution [20]. The reliability of the questionnaire was assessed by means of internal consistency, homogeneity, and test-retest reliability [21,22,23,24]. To analyze the overall internal consistency of the questionnaire and its different dimensions, the McDonald’s omega coefficient was used [21,22]. Homogeneity was assessed by item-total and corrected item-total correlations using Spearman’s correlation coefficient [23]. The overall test-retest reliability of the questionnaire and its dimensions was assessed by repeating the questionnaire one week later with a sample of 30 students using Spearman’s correlation coefficient [24].

To analyze the construct validity of the questionnaire, an exploratory factor analysis (EFA) and a confirmatory factor analysis (CFA) were performed [25,26,27,28,29]. To assess the suitability of the data for EFA, Bartlett’s sphericity tests and the Kaiser–Meyer–Olkin (KMO) test were used. To extract the factors or dimensions of the questionnaire, a principal component analysis with orthogonal rotation was performed using the varimax method [25,27,28].

In the CFA, the first-order models present in the literature were tested: a one-factor model of global cohesion, a cohesion model with two factors representing attraction and integration, a cohesion model with two factors representing social and task aspects, and the cohesion model with four factors, ATG-S, ATG-T, GI-S, and GI-T [12,13,14,15]. The robust weighted least squares mean and variance adjusted (WLSMV) estimator and goodness-of-fit indices were used for the analysis of the fit of the proposed models: Chi-square/degrees of freedom (χ^2^/df), root mean square error of approximation (RMSEA), standardized root mean square residual (SRMR), comparative fit index (CFI), Tucker–Lewis index (TLI), normed fit index (NFI), parsimony ratio (PRATIO), parsimony normed fit index (PNFI), Parsimonic goodness of fit index (PGFI), and Akaike information criterion (AIC) [26,27,28,29].

## 3. Results

### 3.1. Sociodemographic Data

The study sample consisted of 311 students in the last year of the nursing degree; 188 students were from the UCAM and 123 students were from the UMU. The age of the students participating in this research had a mode of 21 years, a mean of 23.54, and a standard deviation of 5.318 and did not follow a normal distribution. The minimum age was 20 years and the maximum age was 54 years. Students aged between 21 and 22 years made up 63.65% of participants.

Of the students, 78.14% were women. Most of the students who participated in the study (71.06%) came from the Region of Murcia. Students with no previous degree represented 81.68% of participants. The most frequent previous degree was that of Auxiliary Nursing Care Technician (31.78%). Of the total number of students, 91.00% were not working at the same time as they were studying nursing.

Table 2 describes the inferential statistical analysis performed between the different sociodemographic variables of the study, indicating the statistical test used and the statistical significance obtained. Statistically significant differences were found for students’ age with respect to gender, previous degree, and work activity. Statistically significant differences were observed between the students’ region of origin with respect to the university where they were studying. Statistically significant differences were also observed with respect to the students’ work activity with respect to their previous degree.

### 3.2. Questionnaire Validation

The descriptive analysis of the scores of the responses to the questionnaire items showed a mean between 3.25 and 3.77; a median of 4 except for items 3, 5, and 12, for which the median was 3; and a mode of 4 except for items 3, 5, and 12, for which the mode was 3. The standard deviation was between 1.001 and 1.041, the skewness was between −0.223 and −0.499, and the kurtosis was between 0.004 and −0.499. The questionnaire items scores did not follow normal distribution as the means, medians, and modes were unequal (Table 3).

In the analysis of the internal consistency of the GEQ and its four dimensions, values above 0.8 were obtained (Table 4). The item-total and corrected item-total correlation coefficients for the questionnaire as a whole and for the four dimensions of group cohesion were greater than 0.44 (Table 5 and Table 6). The test-retest reliability showed correlation values between items and between the different dimensions of the questionnaire higher than 0.8 (Table 7 and Table 8).

The values obtained in the KMO test (0.820) and Bartlett’s sphericity tests (*p* = 0.000) indicated that a factor analysis can be performed on this questionnaire. The extraction by means of the principal component analysis method has allowed us to obtain a communality of the items with values higher than 0.6 (Table 9).

Considering the total variance explained, it can be deduced that the questionnaire was composed of four factors or dimensions that presented eigenvalues greater than 1, which individually explained more than 5% of the variance of the questionnaire and which together accounted for 74.484% of the variance (Table 10). The factor analysis performed by principal component analysis with varimax rotation identified that factor 1 consisted of items 2, 4, and 6, factor 2 included items 1, 3, and 5, factor 3 consisted of items 8, 10, and 12, and factor 4 included items 7, 9, and 11 (Table 11). Table 12 shows the values obtained in the goodness-of-fit indices of the four models of group cohesion tested by confirmatory factor analysis. The cohesion model with four factors is the one that presented the best goodness-of-fit indices (Table 13).

## 4. Discussion

The sample of 311 participants can be considered adequate to perform a rigorous psychometric analysis and adequately establish the validity of GEQ for nursing students undergoing clinical simulation training. The literature suggests that in order to perform a rigorous psychometric analysis and adequately establish the validity of a questionnaire, it is necessary to have a minimum sample of 10 participants per item or at least a total of 300 participants [30].

All the items of the questionnaire presented a standard deviation greater than 1, so that their variability can be considered sufficient to find the differences in group cohesion existing among the participating students. In the Likert-type scales with five response options, a standard deviation of the items between 1 and 1.5 can be considered sufficiently discriminative [31]. The questionnaire items presented negative skewness values of less than 0.5, so it could be deduced that the responses to the items were not biased towards either end of the scale. With values of asymmetry greater than ±1, the item is no longer adequate, and it is necessary to review the wording of the item or even to evaluate its elimination [32,33].

The results obtained by means of the McDonald’s omega test indicated that the questionnaire presented a good internal consistency by presenting values for the questionnaire as a whole and for the different dimensions greater than 0.8. These internal consistency results are superior to those obtained by the validation of this version in Spanish athletes [14] or in work groups of Spanish university students [15]. Both studies used Cronbach’s Alpha coefficient to assess internal consistency. The McDonald’s omega coefficient is more suitable than Cronbach’s alpha coefficient for analyzing the internal consistency of this questionnaire because it is an ordinal response scale with fewer than six options [21,22].

The item-total and corrected item-total correlations showed values greater than 0.44, indicating that the items of the questionnaire analyzed had a good level of homogeneity among them [23,34]. Homogeneity was assessed by the item-total and corrected item-total correlation using Spearman’s correlation coefficient, since these were ordinal qualitative variables [23].

The test-retest reliability showed correlation values between the items and between the dimensions of the questionnaire above 0.8, so that it can be considered that there is excellent agreement between the two completions carried out by the same participating subjects [35]. Spearman’s correlation test was chosen instead of the intraclass correlation coefficient because these are ordinal qualitative variables that do not meet the assumptions of randomness and normality [36].

The results obtained in the KMO test (0.820) and Bartlett’s test of sphericity (*p* = 0.000) indicated the existence of a correlation between the items suitable for factor analysis. A significant Bartlett’s test of sphericity (*p* < 0.05) indicates that the questionnaire items are correlated. A KMO test with a value greater than 0.8 reports a degree of correlation between items suitable for factor analysis [25,27,28]. The communality values of the items were higher than 0.6, indicating the existence of a high proportion of variability that could be explained by the common factors identified in the EFA. These values guarantee the robustness and internal consistency of the questionnaire in measuring its different dimensions [28,37]

Principal component analysis identified that the questionnaire had four factors with eigenvalues greater than 1, which individually explained at least 5% of the variance and together constituted more than 50% of the variance [25,27,28]. The individual factors explained percentages of variance greater than 8% and together constituted 74.484% of the total variance of the questionnaire. This result indicates that the identified factors are statistically sound and adequately represented the underlying structure of the instrument, allowing a reliable and consistent interpretation of the measurable dimensions of the questionnaire. The factors obtained coincide with those observed in the original version of the GEQ [12] and to the short versions of this questionnaire adapted and validated for Spanish professional athletes and Spanish university work groups [14,15].

The EFA by means of principal component analysis with varimax rotation identified that factor 1-ATG-T was made up of items 2, 4, and 6, that factor 2-ATG-S was composed of items 1, 3, and 5, that factor 3-GI-T was made up of items 8, 10, and 12, and that factor 4-GI-S included items 7, 9, and 11. Therefore, it can be seen that the Spanish short version of the GEQ applied to nursing students undergoing clinical simulation training has been fully adjusted to the factor structure composed of four dimensions with three items each proposed by the original study [12] and by the short versions of this questionnaire adapted and validated in Spanish [14,15].

In the CFA, the first-order models present in the literature were tested: a one-factor model of global cohesion, a cohesion model with two factors representing attraction and integration, a cohesion model with two factors representing social and task aspects, and the cohesion model with four factors, ATG-S, ATG-T, GI-S, and GI-T [12,13,14,15]. The results obtained indicated that the first-order four-factor model was the one that presented optimal values in all the goodness-of-fit indices analyzed [26,27,28,29]. These results coincide with those obtained in the original study [12] and with the short versions of this questionnaire adapted and validated in Spanish [14,15]. The findings obtained by CFA have revealed that the four factors obtained by EFA would consistently explain the theoretical hypothesis of group cohesion proposed [7,8,12,13,14,15]. With the validation of this questionnaire, we may have a reliable, valid, and easy-to-complete instrument that would allow us to analyze the group cohesion of Spanish student teams and healthcare professionals.

The choice of participants by nonrandomized convenience sampling and their belonging to two universities in the same geographical area may limit the external validity of this study. Therefore, it would be necessary to confirm the results obtained by randomly selecting nursing students from different geographical areas. Another limitation of the study is that the validation was performed exclusively with nursing students and not with practicing professionals. This could affect the generalizability of the results, as the experiences and perceptions of students may differ to some extent from those of professionals in the field. In addition, it would be desirable to extend this study to other healthcare professionals and to conduct qualitative studies with in-depth interviews to help understand the specific characteristics of group cohesion in healthcare teams.

## 5. Conclusions

The Spanish short version of the Group Environment Questionnaire (GEQ) is a valid and reliable instrument for analyzing group cohesion in teams of nursing students undergoing clinical simulation training. Knowledge of group cohesion in teams formed by nursing students or professionals has important practical implications because it would allow the development of intervention programs to improve interpersonal relationships and collaborative work, increasing the efficiency and quality of healthcare.

## Figures and Tables

**Table 1 nursrep-15-00154-t001:** Spanish short version of the GEQ for nursing students.

Number	Item
1	I like to participate in extracurricular activities with the other members of my group (dinners, excursions…)
2	I am happy with my contributions to the work of the group
3	I have good friends in this group
4	In this group I can perform to the best of my ability
5	Group members are one of the most important social groups to which I belong
6	I like the style of work of this group
7	Group members like to party together
8	Group members join forces to achieve the objectives during the preparation and conduct of the simulation sessions
9	Group members would like to get together a few times after the clinical simulation is over
10	All members take responsibility for a poor group performance
11	Our group members would like to meet in situations other than preparing and conducting simulation sessions
12	If there is a problem during the preparation of the simulation sessions, all members join forces to overcome it

**Table 2 nursrep-15-00154-t002:** Inferential statistics of sociodemographic data.

Variables	Statistical Test	Statistical Significance
University and age	Mann–Whitney U	0.081
University and gender	Chi-square	0.976
University and region of origin	Chi-square	0.000
University and previous degree	Chi-square	0.297
University and work activity	Chi-square	0.435
Gender and age	Mann–Whitney U	0.001
Gender and region of origin	Chi-square	0.989
Gender and previous degree	Chi-square	0.623
Gender and work activity	Chi-square	0.368
Region of origin and age	Kruskal–Wallis	0.427
Region of origin and previous degree	Chi-square	0.630
Region of origin and work activity	Chi-square	0.793
Previous degree and age	Mann–Whitney U	0.000
Previous degree and work activity	Chi-square	0.000
Work activity and age	Mann–Whitney U	0.000

**Table 3 nursrep-15-00154-t003:** Descriptive statistics of the items.

	Item 1	Item 2	Item 3	Item 4	Item 5	Item 6	Item 7	Item 8	Item 9	Item 10	Item 11	Item 12
Mean	3.68	3.70	3.63	3.77	3.36	3.65	3.25	3.60	3.60	3.68	3.63	3.44
Median	4.00	4.00	4.00	4.00	3.00	4.00	3.00	4.00	4.00	4.00	4.00	3.00
Mode	4	4	3	4	3	4	4	4	4	4	4	3
Standard deviation	1.013	1.008	1.001	1.002	1.032	1.001	1.027	1.039	1.001	1.041	1.001	1.017
Skewness	−0.485	−0.475	−0.294	−0.471	−0.223	−0.445	−0.291	−0.489	−0.427	−0.491	−0.499	−0.329
Kurtosis	−0.407	−0.458	−0.491	−0.499	−0.492	−0.315	−0.492	−0.196	−0.217	−0.202	0.004	−0.239

**Table 4 nursrep-15-00154-t004:** Internal consistency of the questionnaire and its dimensions.

	GLOBAL	ATG-S	ATG-T	GI-S	GI-T
McDonald’s Omega	0.855	0.861	0.838	0.805	0.803

**Table 5 nursrep-15-00154-t005:** Item-total correlation coefficients by items.

Item	Item-Total	Corrected Item-Total
1	0.625 *	0.534 *
2	0.706 *	0.627 *
3	0.594 *	0.500 *
4	0.643 *	0.554 *
5	0.649 *	0.562 *
6	0.627 *	0.533 *
7	0.585 *	0.485 *
8	0.572 *	0.468 *
9	0.559 *	0.463 *
10	0.554 *	0.453 *
11	0.562 *	0.463 *
12	0.558 *	0.445 *

Note: * The correlation is significant at the 0.01 level (bilateral).

**Table 6 nursrep-15-00154-t006:** Item-total correlation coefficient by dimensions.

Dimension		ATG-S			ATG-T			GI-S			GI-T	
	Item 1	Item 3	Item 5	Item 2	Item 4	Item 6	Item 7	Item 9	Item 11	Item 8	Item 10	Item 12
Item-total	0.821 *	0.896 *	0.897 *	0.887 *	0.873 *	0.819 *	0.806 *	0.829 *	0.849 *	0.837 *	0.837 *	0.788 *
Corrected item-total	0.617 *	0.755 *	0.768 *	0.752 *	0.725 *	0.595 *	0.563 *	0.633 *	0.648 *	0.616 *	0.633 *	0.557 *

Note. Statistical test used: Spearman’s correlation coefficient. * The correlation is significant at the 0.01 level (bilateral).

**Table 7 nursrep-15-00154-t007:** Test-retest reliability of the items.

Test/Retest	Item 1	Item 2	Item 3	Item 4	Item 5	Item 6	Item 7	Item 8	Item 9	Item 10	Item 11	Item 12
Item 1	0.928 *	-	-	-	-	-	-	-	-	-	-	-
Item 2	-	0.840 *	-	-	-	-	-	-	-	-	-	-
Item 3	-	-	0.888 *	-	-	-	-	-	-	-	-	-
Item 4	-	-	-	0.910 *	-	-	-	-	-	-	-	-
Item 5	-	-	-	-	0.852 *	-	-	-	-	-	-	-
Item 6	-	-	-	-	-	0.976 *	-	-	-	-	-	-
Item 7	-	-	-	-	-	-	0.871 *	-	-	-	-	-
Item 8	-	-	-	-	-	-	-	0.818 *	-	-	-	-
Item 9	-	-	-	-	-	-	-	-	0.952 *	-	-	-
Item 10	-	-	-	-	-	-	-	-	-	0.882 *	-	-
Item 11	-	-	-	-	-	-	-	-	-	-	0.846 *	-
Item 12	-	-	-	-	-	-	-	-	-	-	-	0.812 *

Note. Statistical test used: Spearman’s correlation coefficient. * The correlation is significant at the 0.01 level (bilateral).

**Table 8 nursrep-15-00154-t008:** Test-retest reliability of the dimensions.

Test/Retest	ATG-S	ATG-T	GI-S	GI-T
ATG-S	0.948 *	-	-	-
ATG-T	-	0.940 *	-	-
GI-S	-	-	0.897 *	-
GI-T	-	-	-	0.926 *

Note. Statistical test used: Spearman’s correlation coefficient. * The correlation is significant at the 0.01 level (bilateral).

**Table 9 nursrep-15-00154-t009:** EFA. Communalities.

	Initial	Extraction
Item 1	1.000	0.687
Item 2	1.000	0.828
Item 3	1.000	0.854
Item 4	1.000	0.820
Item 5	1.000	0.857
Item 6	1.000	0.679
Item 7	1.000	0.634
Item 8	1.000	0.725
Item 9	1.000	0.770
Item 10	1.000	0.745
Item 11	1.000	0.758
Item 12	1.000	0.681

Note: Extraction method: principal component analysis.

**Table 10 nursrep-15-00154-t010:** EFA. Total variance explained.

Item	Initial Eigenvalues			Sums of Charges Squared by Extraction			Sums of Charges Squared by Rotation		
	Total	% of Variance	% Accumulated	Total	% of Variance	% Accumulated	Total	% of Variance	% Accumulated
1	4.641	38.673	38.673	4.641	38.673	38.673	2.347	19.561	19.561
2	2.043	17.022	55.694	2.043	17.022	55.694	2.255	18.795	38.356
3	1.197	9.973	65.667	1.197	9.973	65.667	2.177	18.145	56.502
4	1.058	8.817	74.484	1.058	8.817	74.484	2.158	17.982	74.484
5	0.617	5.141	79.625						
6	0.499	4.158	83.783						
7	0.462	3.849	87.632						
8	0.377	3.140	90.772						
9	0.346	2.886	93.658						
10	0.337	2.812	96.470						
11	0.227	1.894	98.364						
12	0.196	1.636	100.000						

Note: Extraction method: principal component analysis.

**Table 11 nursrep-15-00154-t011:** EFA. Matrix of rotated components.

	Factor 1 (ATG-T)	Factor 2 (ATG-S)	Factor 3 (GI-T)	Factor 4 (GI-S)
Item 1	0.249	0.693	0.005	0.075
Item 2	0.855	0.238	0.133	0.150
Item 3	0.168	0.901	0.055	0.106
Item 4	0.875	0.188	0.087	0.104
Item 5	0.205	0.881	0.149	0.130
Item 6	0.696	0.213	0.144	0.169
Item 7	0.136	0.128	0.260	0.729
Item 8	0.120	0.106	0.808	0.214
Item 9	0.148	0.080	0.135	0.850
Item 10	0.149	0.095	0.826	0.178
Item 11	0.113	0.084	0.167	0.843
Item 12	0.058	0.003	0.811	0.141

Note: Extraction method: principal component analysis. Rotation method: varimax with Kaiser normalization.

**Table 12 nursrep-15-00154-t012:** Goodness-of-fit indexes of the models of the GEQ.

Models	Absolute Fit Indexes			Incremental Fit Indexes			Parsimony Fit Indexes			
	X^2^/df	RMSEA	SRMR	CFI	TLI	NFI	PRATIO	PNFI	PGFI	AIC
1 factor: Global Cohesion	15.3518	0.2150	0.1340	0.546	0.445	0.532	0.802	0.436	0.447	9820
2 factors: Attraction and Integration	9.5471	0.1660	0.0812	0.735	0.670	0.715	0.803	0.574	0.590	9499
2 factors: Social and Task	12.6415	0.5490	0.1530	0.638	0.549	0.622	0.803	0.449	0.512	9663
4 factors	2.1660	0.0495	0.0453	0.967	0.955	0.941	0.818	0.748	0.803	9107

**Table 13 nursrep-15-00154-t013:** Goodness-of-fit indices of the four-factor model of the GEQ.

Type	Good Fit	Results
Absolute Fit Indices		
Chi-square/degrees of freedom	<5	2.1660
Root Mean Square Error of Approximation (RMSEA)	<0.05	0.0495
Standardized Root Mean Square Residual (SRMR)	<0.05	0.0453
Incremental Fit Indices		
Comparative Fit Index (CFI)	>0.9	0.955
Tucker–Lewis Index (TLI)	>0.9	0.941
Normed Fit Index (NFI)	>0.9	0.967
Parsimonious Fit Indices		
Parsimony ratio (PRATIO)	Proximity to 1	0.818
Parsimony Normed fit Index (PNFI)	Proximity to 1	0.748
Parsimonious Goodness of Fit Index (PGFI)	Proximity to 1	0.803
Akaike Information Criterion (AIC)	Proximity to 0	9107

## Data Availability

The data used to support the findings of this study are available from the corresponding author upon request.

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
