# Peer review of "Validation of the Group Environment Questionnaire (GEQ) in a Simulated Learning Environment"

_nursrep, 2025, doi:10.3390/nursrep15050154_

Round 1
Reviewer 1 Report
Comments and Suggestions for Authors
The abstract is well written and straightforward
Throughout the paper, be sure paragraphs are at least 3 sentences long.
Suggest changing the first sentence to something that makes an impactful statement about the importance of teamwork, learning to work in teams or groups. The third sentence might be better if moved to the first sentence.
Line 36 - suggest revising second sentence to read "These changes in technology and care complexity make it necessary for ....then follow this sentence with a revision of the fourth sentence "Healthcare teams must possess non-technical skills ....."
Has the GEQ been utilized in previous studies? Suggest including summary of studies this questionnaire has been used in and previous reliability and validity if available.
Suggest moving the ethical statement up to maybe after "procedure" section. Suggest adding some information describing the study team/researchers and if there was any potential for conflict of interest of coercion if the researchers were professors/instructors working with the students.
Is there information in the paper about the translation of the instrument from its original version to Spanish? I may have missed it if it is included.
Suggest adding to the conclusion or discussion how this particular questionnaire is important over other questionnaires that study group work and/or teamwork. What are the benefits of this tool? Is it shorter or easier to administer? Does it have better reliability and validity? Is it applicable to a wider variety of situations?
Author Response
Dear reviewer.
I am sending you my rebuttal letter to your comments and suggestions.
Kind regards.

Reviewer 2 Report
Comments and Suggestions for Authors
Please refer to the attached file.

Author Response

(The authors gave the same response as above.)
